# Differences in the Quality of Life of Patients Recently Diagnosed with Crohn’s Disease and Ulcerative Colitis

**DOI:** 10.3390/ijerph20166576

**Published:** 2023-08-14

**Authors:** Purificación Bernabéu Juan, Paula Cabezos Sirvent, Laura Sempere Robles, Ana van-der Hofstadt Gomis, Jesús Rodríguez Marín, Carlos J. van-der Hofstadt Román

**Affiliations:** 1Unidad de Psicología Hospitalaria, Hospital General Universitario de Alicante, C/Pintor Baeza 12, 03010 Alicante, Spain; rod.marin@umh.es; 2Instituto de Investigación Sanitaria y Biomédica de Alicante (ISABIAL), C/Pintor Baeza 12, 03010 Alicante, Spain; lausemro@hotmail.com; 3Departamento de Psicología de la Salud, Universidad Miguel Hernández, Avda de la Universidad s/n Edificio Altamira, 03202 Elche, Spain; paula.cabezos@alu.umh.es (P.C.S.); ana.van-der@goumh.umh.es (A.v.-d.H.G.); 4Servicio de Gastroenterología, Hospital General Universitario de Alicante, C/Pintor Baeza 12, 03010 Alicante, Spain

**Keywords:** inflammatory bowel disease, Crohn’s disease, ulcerative colitis, quality of life, impact, recently diagnosed

## Abstract

Inflammatory bowel diseases (IBD) are chronic diseases, encompassing Crohn’s disease (CD) and ulcerative colitis (UC). An IBD diagnosis has an impact on the quality of life of patients; this impact can be different according to the type of disease. Objective: This study aimed to analyze the differences in the impact on quality of life in the early stages after diagnosis in patients with CD and UC. Patients and methods: This was an observational, multi-center, and cross-sectional study, with the participation of 156 patients recently diagnosed with IBD (<6 months) from 4 hospitals from the Health Council of the Valencian Community. The patients were assessed through the use of the Inflammatory Bowel Disease Questionnaire (IBDQ-32), which measures the quality of life when living with IBD. Results: The sample was composed of 80 patients with CD (51.0%) and 76 patients with a UC diagnosis. The mean age was 42.3 ± 16.2. The CD patients were more affected (42.5%) in their general quality of life than the UC patients (17.1%) (*p* = 0.001). In the dimensions of the IBDQ-32, the patients with CD showed significant differences in the systemic, emotional, and social spheres. The bowel dimension scores were similar in both groups. Conclusions: The patients who were recently diagnosed with CD were more affected regarding their quality of life as compared to those who were diagnosed with UC. Psychological care must be considered to mitigate the impact of an IBD diagnosis.

## 1. Introduction

An inflammatory bowel disease (IBD) diagnosis is a life stressor that can lead to the appearance of emotional symptomatology. Its chronic and unpredictable course, along with its bothersome and painful symptoms, can make patients become worried about certain aspects, such as sphincter control, social isolation, fatigue, the development of cancer, or the need for surgery and can also affect the area of work [1,2].

The months following an IBD diagnosis are particularly complex for patients due to the many tests needed, new medical information, adjustments of medications, and discussions and implementation of different therapeutic strategies [3].

The uncertain and unforeseeable nature of IBD, the possible long-term complications, and the psychological and economic burdens of this disease have negative impacts in different areas and levels, with one of them being quality of life [4].

Diverse studies have shown worse scores in the quality of life of patients who have been recently diagnosed with IBD as compared to those who have lived with it for some time [5,6,7,8,9]. Likewise, there is greater psychiatric morbidity in both ulcerative colitis (UC) and Crohn’s disease (CD), especially during the first year of living with the disease, which is considered a direct consequence of the stress after the diagnosis, as well as the novelty, severity, and variability of the symptoms [10]. In addition, the improvement in the quality of life at the start of treatment can have positive effects on adherence and the prognosis of the disease, so it is necessary to pay close attention to the early pattern of quality of life and its changes in patients with IBD [11].

The clinical characteristics of CD and UC are very different. While the most common manifestation of UC is diarrhea, in CD, it is frequent to find abdominal pain, loss of weight, or anemia; these appear only in the most severe and extensive types of UC [12]. Along the same lines, there is a greater risk of intestinal stenosis and risk of surgery with CD as compared to UC in the case of a late diagnosis [13]. Diverse studies indicate that the systemic dimension of the quality of life, characterized by symptoms such as fatigue, general malaise, and sleep alteration, is the most affected in IBD, and more specifically, the data show a greater systemic impact in CD than in UC [14,15,16].

Patients with CD exhibit greater impacts in the psychological dimension, with subjects who have this disease showing higher levels of anxiety and depression [15,16]. Bernstein et al. [17] point out a greater risk of schizophrenia after being diagnosed with CD as compared to UC. Along the same line, Ananthakrishnan et al. [18] have shown a greater frequency of stressful life events 6 months before a diagnosis of CD as compared to UC. Lastly, other studies have observed a greater incidence and risk of suicide in patients with CD as compared to UC [10,19].

With respect to quality of life, different studies have shown that patients with CD make worse assessments of their quality of life as compared to patients with UC [5,8,15,16,20], although the number of studies that have investigated quality of life just after patients received a diagnosis of CD or UC are scarce, which results in a lack of information in this field. Thus, the aim of the present study is to assess the association of IBD with the quality of life of patients who have been recently diagnosed and to compare this effect on patients with CD and UC at the time of diagnosis in order to increase our knowledge about the magnitude of this problem.

## 2. Method

### 2.1. Participants

A total of 156 patients (87 men) participated in the study. These individuals were diagnosed with CD or UC for less than or equal to 6 months before the assessment. They were provided with care at the digestive (gastroenterology) units at 4 public hospitals located in the Valencian Community: General University Hospital Dr. Balmis, University Hospital San Juan of Alicante, General University Hospital of Elche, and Polytechnic University Hospital La Fe.

The sample was composed of 80 patients with CD (51%) and 76 patients with UC. The inclusion criteria were: being 18 years old or older; having a diagnosis of IBD according to the European Crohn’s and Colitis Organization, with a diagnosis date equal to or less than 6 months before the assessment; signing an informed consent form and agreeing to participate in the assessment process; and having adequate reading and writing skills and autonomy to be able to complete the evaluation questionnaire. The exclusion criteria were: having severe mental health problems or being in an acute disease phase, the characteristics of which could have repercussions on the diagnostic and therapeutic process.

A consecutive sampling method was used to recruit study participants in order to include all accessible subjects who met the proposed inclusion criteria.

### 2.2. Variables and Instruments

#### 2.2.1. Clinical and Sociodemographic Variables

The variables included were the type of disease (CD or UC), diagnosis date, age at the time of the study, sex (male or female), level of education (non-university or university), marital status at the time of the study (married, separated/divorced, single, or widowed) and employment (employed, unemployed, student, retired, disabled, temporary incapacity to work, or homemaker).

#### 2.2.2. Dependent Variable

The dependent variable was the effect on quality of life, as measured by the Inflammatory Bowel Disease Questionnaire (IBDQ-32). The version adapted and validated in Spanish by Masachs et al. [21] was used. The IBDQ-32 is composed of 32 items and collects information about the quality of life of the patient with 4 dimensions: bowel symptoms (10 items), emotional function (12 items), social function (5 items), and systemic symptoms (5 items). Each item is scored with a 7-point Likert scale, with 7 being the highest score for quality of life and 1 being the lowest, resulting in a possible range of 32 to 224 points for general quality of life [16].

The magnitude of the impact of the disease on each dimension was taken into account in order to assess the individual dimensions; this was done by considering an inverse score with respect to quality of life [22]. Thus, for the bowel dimension, a score ranging from 10 to 29 indicated a high impact; a score between 30 and 49 indicated a medium impact, and a score between 50 and 70 indicated a low impact. For the systemic dimension, a score between 5 and 14 indicated a high impact; a score between 15 and 24 indicated a medium impact, and a score between 25 and 35 indicated a low impact. A high impact in the emotional dimension was represented by a score between 12 and 35; a medium impact in this domain was indicated by a score between 36 and 59, and a low impact was indicated by a score between 60 and 84. Finally, a high impact in the social dimension was indicated by a score between 5 and 14; a medium impact in this domain was indicated by a score between 15 and 24, and a low impact was indicated by a score between 25 and 35 [22].

In this study, due to the sample size and the interest in observing if an effect was present or not, the high- and medium-impact groups were grouped together to decrease the data dispersion and to concentrate the information obtained.

### 2.3. Procedure

The study was a multi-center, observational, descriptive, and cross-sectional study.

Data collection was conducted between January 2018 and March 2020 and was interrupted due to the state of emergency declared due to COVID-19 in March. The sample was accessed through the digital medical histories of each hospital thanks to the System for Health Management from the Health Council (ABUCASIS). Thus, considering the list of patients from each hospital with a recent diagnosis of IBD, the potential participants were first contacted by phone and were then given an appointment for an in-person interview, which coincided with a follow-up appointment at the hospital. The assessment was performed by psychologists.

After being informed about the study, the patients who decided to participate signed an informed consent form prior to their participation in the study. The present study was framed within a larger project that was approved by the Ethics Committee of Research from the General University Hospital Dr. Balmis (CEIm), dated 17 May 2018 (Ref. CEI PI2018/047).

This study was conducted according to Law 17/2007, in effect from 3 July, on Biomedical Research and the main principles of the Declaration of Helsinki (2013). The treatment, communication, and cessation of use of the personal data of the study participants complied with Organic Law 15/1999 on the Protection of Personal Data.

### 2.4. Analysis of Data

Descriptive statistics were utilized to describe the sociodemographic and clinical characteristics of the sample, including frequencies with percentages, means, and standard deviations (SD) (the last two were used for the age variable). First, Student’s *t*-test was used to analyze the quantitative variables with a parametric distribution. Then, to analyze the correlation between the qualitative variables, Pearson’s chi-square test was utilized. Next, a confidence level of 95% was used. Finally, the SPSS version 25.0 statistical package (Chicago, IL, USA). was utilized for the analyses.

## 3. Results

A descriptive analysis of the sociodemographic and clinical characteristics of the study sample was performed, which included the following variables: age (mean and standard deviation), sex, clinical diagnosis, level of education, marital status, and employment.

Table 1 shows the sociodemographic and clinical variables of the subjects who participated in the study.

Figure 1 shows the percentage of the sample reporting an impact on their general quality of life, separated according to diagnosis. While only 17.1% of the recently-diagnosed patients with UC indicated an effect on their general quality of life, 42.5% of those diagnosed with CD were affected. The Chi-squared test (*p* = 0.001) indicated a statistically significant difference in the impact of these two diagnoses.

Table 2 describes how the dimensions of the psychological variable studied were affected, again according to the presence of a CD or UC diagnosis. The results from this table were found to be along the same line as the results of the effect on general quality of life.

As observed, the dimension with the greatest number of affected individuals, for both CD and UC, was the systemic dimension, with 72 subjects (46.1%), followed by the emotional dimension, with 54 (34.6%) patients; the bowel dimension, with 41 (26.3%) patients; and the social dimension, with 22 patients (14.1%).

## 4. Discussion

The aim of the present study was to analyze the association of IBD with the quality of life of patients who were diagnosed with CD or UC within a period of time equal to or less than 6 months before the assessment with the IBDQ-32 instrument.

After analyzing the results obtained, a statistically significant difference in the quality of life impact was confirmed for the subjects who were recently diagnosed with CD as compared to the subjects diagnosed with UC, with those with CD being the most affected.

Few studies were found that compared the level of quality of life of patients with CD and UC at the moment of the diagnosis, and they showed a lack of statistical significance in their results. McCombie et al. [9] conducted a study with a sample diagnosed with IBD within 6 months before the assessment without statistically significant differences observed in the quality of life of patients with CD and UC. Burisch et al. [5], in their ecological study EpiCom on the incidence of IBD in Europe, analyzed the quality of life of 1079 patients with IBD at the moment of diagnosis and after a year of living with the disease, observing a lower general quality of life and in different dimensions in their results in patients who were recently diagnosed with CD as compared with UC. Nevertheless, this study did not find statistically significant differences between these two diagnoses, as their objective was to analyze the regional differences between Eastern and Western Europe on the quality of life at diagnosis and after a year of treatment. Jaghult et al. [8] observed a higher quality of life in patients with UC as compared with CD patients, although these differences were not significant, and the sample was composed of patients with IBD with a duration of less than 2 years. The results from Andrzejewska et al. [23] were not significant either when the authors compared the quality of life of patients with CD and UC, showing only a total of 28.8% of the sample who were diagnosed within a period of two years before the assessment. López et al. [20] concluded that their group of subjects with CD at the time of diagnosis had a similar general quality of life as compared with their subjects with UC.

In the studies in which the time of diagnosis was not a study variable, the results were disparate. While some of the studies pointed to a significant difference in the quality of life between UC and CD, with the latter being the most affected [6,7,15,16], others did not show significant differences between the UC and CD patients, despite the CD subjects obtaining worse scores in the quality of life as compared to the UC patients [24,25,26]. In the studies by Caplan et al. and Ganguli et al. [27,28], the results obtained showed a higher quality of life in the group of subjects with CD, although none of these studies showed statistically significant differences between the groups.

As for the dimensions of quality of life, the results indicated, on the one hand, an impact in both the patients who suffered from CD and those who suffered from UC in all 4 dimensions, and on the other hand, significant differences were seen in the emotional, systemic, and social dimensions, with the CD subjects being more affected as compared with the UC subjects. However, the results obtained were not in agreement with previous studies, such as Huamán et al. [25], which did not find significant differences between the CD and UC subjects or in the quality of life dimensions, or Haapamaki et al. [6,7], which found statistically significant differences only in the social and bowel dimensions.

The differences in the results of the present study and those mentioned above could be due to the particular nature of the sample, which included subjects who had been recently diagnosed (less than 6 months), while in the other studies mentioned, the participants had not been recently diagnosed. In light of these data, future studies could analyze changes in the impact of these diseases on the quality of life dimensions of patients in a longitudinal manner starting from the time the diagnosis was made.

Along the same line as in the previous literature, in this study, the dimension that was most affected in both IBD groups was the systemic dimension, which is characterized by symptoms such as fatigue, sleep alteration, weight loss, and general malaise [5,16,20,26]. The other dimensions that were greatly affected in both groups were the emotional, bowel, and social dimensions. These results differed from those obtained in the studies by Burisch et al. [5], Cohen et al. [14], Huamán et al. [16], and López et al. [25], who were in agreement that the most affected dimension after the systemic dimension was the social dimension. In the study conducted by Cohen et al. [14], in which the sample was composed of newly diagnosed IBD patients, a significant correlation was observed between greater systemic symptoms, such as fatigue, and a worse general quality of life, which supports the relevance of this symptom as the most affected in this study and the other ones cited. It must be underlined that in the studies described, the bowel dimension was less affected as compared to the systemic, emotional, or social dimensions.

Thus, there is a lack of an association between disease activity and a worse quality of life, underlining the greater importance of the other dimensions that differ from the disease’s digestive symptoms. A possible explanation for these data could be the weight of the consequences of the disease on the everyday life of a person, taking into account both the systemic symptoms described as well as the psychological and social symptoms associated with the change in lifestyle and the self-concept of the subjects who have been recently diagnosed with IBD.

Burisch et al. [5], in a study of the incidence of IBD in Europe, highlighted their observation of better medical care and more information with respect to the digestive symptoms, as compared to the other dimensions, such as the emotional and social dimensions, and symptoms related with the disease. Along the same line, considering the results from the present and other similar studies, the comparable importance of the role of information and support, both medical and psychological, can be affirmed in patients with IBD.

The results described in the present study and in the literature reviewed present a complex situation for individuals who have been recently diagnosed with IBD, as they are characterized by greater psychological vulnerability and stress due to the appearance of this chronic disease.

Based on the new concerns and needs that emerge after a diagnosis of IBD, the comprehensive care of these patients becomes a priority. In their study, Bernstein et al. [17] showed the impact of greater knowledge of the disease on the level of quality of life of patients who had been recently diagnosed with IBD. Lores et al. [29], in their study, observed that the patients with IBD who accepted a psychological intervention improved their levels of anxiety, depression, stress, and quality of life. Along the same line, the European Crohn’s and Colitis Organization (ECCO) [30] created a document on the needs of IBD patients with respect to the quality of care received, showing that two of the most significant factors that had the greatest impact on the improvement of quality of life and the relationship between physicians and patients were adequate education about the disease and the joint treatment of anxiety and depression. Thus, it can be concluded that there is ample support based on the scientific literature for working on IBD from a systemic approach, bearing in mind the comorbidities that accompany this diagnosis.

The strengths of the study come from being one of the few that analyzed the impact of IBD on the quality of life and the differences of this impact between CD and UC in a period of time equal to or less than 6 months from the diagnosis. Some described studies, such as those by Andrzejewska et al., Engel et al., and Jaghult et al. [3,8,23], underlined their use of a sample or part of a sample with recently diagnosed IBD, although these samples participated in the study after being diagnosed within a period of time of 2 to 5 years before the study.

As for the limitations of the present study, the first limitation is the reduced size of the sample when separating the total sample into two diseases. The second limitation is geographical; even though this study was a multi-center study, the sample exclusively came from Valencian Community. Lastly, the ending of the study due to the COVID-19 state of emergency is also considered a limitation.

## 5. Conclusions

Based on the results of the present study, it can be concluded that there is a significant difference in disease impact on the quality of life of individuals who were recently diagnosed with CD as compared to those who were diagnosed with UC, and this effect is effectively due to the disease and not to chance. The importance of this difference comes from the acquisition of knowledge about the repercussions of CD beyond its clinical diagnosis. For this, it is also pertinent as a future line of research to assess and implement other types of interventions in digestive health that prioritize the assessment of different patient needs in order to be able to detect them and intervene from the moment a diagnosis is made. This must occur considering the psychological repercussion that a CD entails and while providing skills and strategies that allow the CD patient to be able to effectively manage their disease and have a better quality of life at the psychological and social levels during this first important period. In this sense, one implication of this study would be to propose the integration of health psychology professionals in multidisciplinary care teams, thus becoming able to provide patients the support that is necessary in each case from the first moment when they come to the doctor to after their diagnosis to reduce the impact that their diagnosis has on their quality of life, especially in the case of CD patients due to the increased impact they experience.

## Figures and Tables

**Figure 1 ijerph-20-06576-f001:**
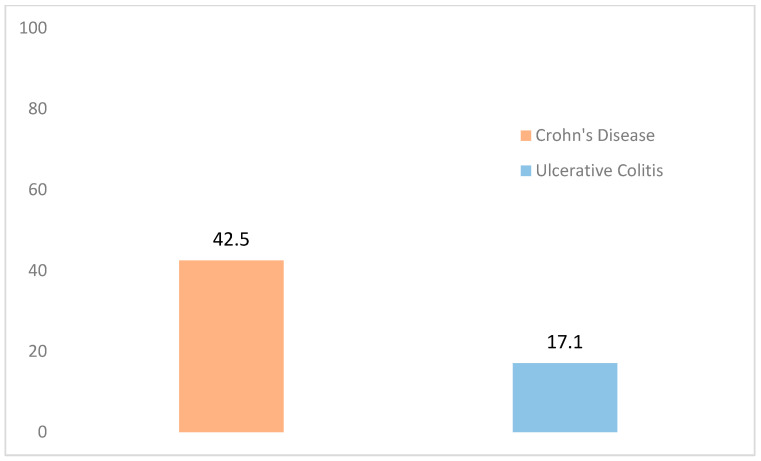
General Quality of Life of the Sample According to Diagnosis (Impact). Note. *p* = 0.001.

**Table 1 ijerph-20-06576-t001:** Clinical and Sociodemographic Variables (*n* = 156).

Clinical and Sociodemographic Variables	*n* (%)
*Age* (mean ± SD)	42.3 ± 16.2
*Sex*	
Male	87 (55.8)
Female	69 (44.2)
*Clinical Diagnosis*	
Crohn’s Disease	80 (51.0)
Ulcerative Colitis	76 (49.0)
*Education*	
Non-university Education	107 (68.6)
University Education	49 (31.4)
*Marital Status*	
Married/Living with a partner	94 (60.2)
Separated/Divorced	12 (7.7)
Single	47 (30.1)
Widower	3 (2.0)
*Employment*	
Employed	83 (53.2)
Unemployed	23 (14.7)
Student	17 (10.9)
Retired	13 (8.3)
Disabled	2 (1.3)
Temporary incapacity to work	7 (4.5)
Homemaker	11 (7.1)

**Table 2 ijerph-20-06576-t002:** Quality of Life of the Sample According to Diagnosis. IBDQ Dimensions (Impact).

IBDQ Dimensions	CD (*n* = 80) *n* (%)	UC (*n* = 76) *n* (%)	*p*
Systemic	47 (58.8)	25 (32.9)	0.001 *
Social	17 (21.3)	5 (6.6)	0.008 *
Emotional	40 (50.0)	14 (18.4)	˂0.001 *
Bowel	26 (32.6)	15 (19.7)	0.070

Note. IBDQ: Quality of Life Questionnaire Specific to IBD, * *p ˂* 0.05.

## Data Availability

The data are not publicly available due to ethical reasons.

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
