# Peer review of "Differences in the Quality of Life of Patients Recently Diagnosed with Crohn’s Disease and Ulcerative Colitis"

_ijerph, 2023, doi:10.3390/ijerph20166576_

Round 1
Reviewer 1 Report
see attached file

Reviewer 2 Report
For the authors,
I congratulate the authors for their work. The subject is of great interest and relevance since inflammatory bowel disease is responsible for strongly affecting the psychological health of individuals.
The introduction is clear, concrete, and objective. The objective of the work is clearly defined.
The methodology and results are well described.
The discussion fully integrates existing studies with data from the present study.
The authors identify the limitations and the added value of the study.
However, I have some suggestions to enrich the work. I hope you are available to add these changes.
In the instrument description section, please add examples of items on the scales of the quality-of-life questionnaire. Although the three scales are intuitive, the “systemic symptoms” scale may generate some curiosity. It would be better to list one or two items and explain what each scale measures.
I think the article could have more details about the quality of life since this is the core (and only objective) of the study. Specifically, I suggest adding differences in quality of life according to sociodemographic characteristics (variables in Table 1). If possible, include clinical variables relevant to quality of life. These analyses will complement the work and enlighten readers about who are the “groups” of people at risk of having a worse quality of life. For designing and implementing interventions, these data are essential.
July 16, 2023
Round 2
Reviewer 1 Report
please see attached file
